

# Boundary condition and reflection anomaly in 2 + 1 dimensions

**Jiunn-Wei Chen[1,2,3⋆], Chang-Tse Hsieh[1,3†] and Ryutaro Matsudo[1‡]**

**1** Department of Physics and Center for Theoretical Physics,
National Taiwan University, Taipei 10617, Taiwan
**2** Leung Center for Cosmology and Particle Astrophysics,
National Taiwan University, Taipei 10617, Taiwan
**3** Physics Division, National Center for Theoretical Sciences, Taipei 10617

⋆ jwc@phys.ntu.edu.tw , † cthsieh@phys.ntu.edu.tw , ‡ matsudo@phys.ntu.edu.tw

## Abstract

It is known that the 2 + 1d single Majorana fermion theory has an anomaly of the reflection, which is canceled out when 16 copies of the theory are combined. Therefore, it is expected that the reflection symmetric boundary condition is impossible for one Majorana fermion, but possible for 16 Majorana fermions. In this paper, we consider a reflection symmetric boundary condition that varies at a single point, and find that there is a problem with one Majorana fermion. The problem is the absence of a corresponding outgoing wave to a specific incoming wave into the boundary, which leads to the non-conservation of the energy. For 16 Majorana fermions, it is possible to connect every incoming wave to an outgoing wave without breaking the reflection symmetry. In addition, we discuss the connection with the fermion-monopole scattering in 3 + 1 dimensions.

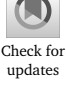

# 1  Introduction

At the quantum level, there can be an obstruction of gauging a symmetry, which is called an 't Hooft anomaly [1]. Clarifying whether the symmetry is anomalous is important because an anomaly provides a powerful constraint on the phase of the theory. The theory with an anomaly in $d$ dimensions can also be employed for the classification of symmetry-protected topological (SPT) phases in $d+1$ dimensions [2–5]. This application is based on the conjecture that when we define an SPT phase with the symmetry $G$ on a manifold with a boundary, it is always associated with boundary degrees of freedom that carry an anomaly of the symmetry $G$, but the total system does not have the anomaly.

The classification of 3+1 dimensional SPT phases was argued for several symmetries [4–7]. Remarkably, the free fermion classification of SPT phases with the reflection symmetry, given by $\mathbb{Z}$, is reduced to $\mathbb{Z}_{16}$ when interactions are introduced [6]. This classification is confirmed by calculating the eta invariant, which is the action of the free fermion SPT phase [8]. The surface theory of this SPT phase is the $2+1$d free massless Majorana fermion theory, which has an anomaly of the reflection symmetry. Corresponding to this relation, the 16 Majorana fermion theory does not have the reflection anomaly [8–10]. In $3+1$ dimensions, the classification of the $\mathbb{Z}_4$ symmetry, with its generator $X$ satisfying $X^2 = (-1)^F$, is also given by $\mathbb{Z}_{16}$ [9]. This coincidence is explained by an isomorphism between corresponding bordism groups [11]. The $3+1$d single Majorana fermion theory has the anomaly of the $\mathbb{Z}_4$ symmetry, and the 16 Majorana fermion theory does not have the anomaly [11].

The relation between a bulk SPT phase and a surface anomalous theory leads to another characteristic of anomalies, non-edgeablity [2, 12, 13], which means the absence of boundary conditions preserving the corresponding symmetry. In $1+1$ dimensions, the existence of boundary conditions preserving non-anomalous symmetries and the absence of boundary conditions preserving anomalous symmetries were confirmed for perturbative anomalies [14] and certain discrete symmetries, including time-reversal, in specific CFTs [12, 13, 15]. Such studies in $1+1$d CFTs are possible because the language of boundary conformal field theory (BCFT) describes all possible boundary conditions. There are non-trivial boundary conditions that can only be described using the language of BCFT and cannot be expressed as a linear equation of the fields at the boundary. For example, the chiral fermion parity in the massless free Majorana fermion theory whose flavor number is a multiple of eight is only preserved by boundary conditions of this kind [12, 15]. However, our current understanding of anomalous theories on a manifold with a boundary in higher dimensions remains limited.

In this paper, we examine the possibility of imposing a boundary condition preserving several symmetries including the reflection in $2+1$ dimensions to check the equivalence of the non-edgeability and the anomaly.

In Sec. 2, we confirm that simple linear boundary conditions do not preserve the anomalous symmetries listed in Tab. 1. We also show that several symmetries without anomalies can be preserved by linear boundary conditions, but the reflection symmetry cannot be, even when it is non-anomalous.

Table 1: The classification of the anomalies in $2+1$ dimensions. Actions of the symmetries to a Dirac fermion are written in Eqs. (5)-(10). The classification is given, e.g., in Ref. [5,7,17] as the classification of fermionic SPT phases in $3+1$ dimensions.

| Symmetries | $(-1)^F$ | $R$ | $\tilde{R}$ | $U(1)$ | $U(1) \times R$ | $U(1) \rtimes CR$ | $U(1) \rtimes \widetilde{CR}$ |
|---|---|---|---|---|---|---|---|
| Anomalies in 3d | 0 | $\mathbb{Z}_{16}$ | 0 | 0 | $\mathbb{Z}_8 \times \mathbb{Z}_2$ | $\mathbb{Z}_2{}^3$ | $\mathbb{Z}_2$ |

In Sec. 3, we impose a boundary condition that varies at a single point, taken as the origin, in order to maintain the reflection symmetry in the single Majorana fermion theory. We show that, with this boundary condition, there is no corresponding outgoing wave for a specific incoming wave into the origin. This means that the incoming wave must vanish at the origin, leading to a violation of energy conservation. To obtain a corresponding outgoing wave, it is necessary to introduce an additional Majorana fermion and impose an extra boundary condition at the origin. We show that this additional boundary condition breaks the reflection symmetry.

In Sec. 4, we consider the possibility of boundary conditions preserving the reflection symmetry in the presence of $N_f$ Majorana fermions. By restricting the fermion fields to the components without corresponding outgoing or incoming waves, the system is reduced to $1+1$ dimensional one on the half line parametrized by the radial distance. The boundary condition at the origin corresponds to the boundary condition of the $1+1$d fermions. The reflection symmetry in the original $2+1$d theory is reduced to the chiral fermion parity in the $1+1$d theory. By using the known facts of this $1+1$d symmetry, we show that it is possible to impose a boundary condition preserving the reflection symmetry when the number $N_f$ of the Majorana fermions is a multiple of 16. This is consistent with the absence of the reflection anomaly in such cases.

In Sec. 5, we discuss the relation between the reflection anomaly in $2+1$ dimensions and the fermion-monopole scattering in $3+1$ dimensions. Also in the monopole background, the absence of the corresponding outgoing wave for a specific incoming wave is observed when the theory suffers from the mixed gauge-gravitational anomaly. We can explicitly relate this problem to the problem in the $2+1$d setup by restricting the 4d fermion to the modes in which the third component of the angular momentum is zero.

In Sec. 6, we discuss the $3+1$d anomalies of $\mathbb{Z}_4$ and $U(1) \times \mathbb{Z}_2$ that can be derived from the $2+1$d anomaly of $R$ and $U(1) \times R$ respectively. In the same way as the $2+1$d setup, we impose a boundary condition changing at a line to preserve the symmetry. By restricting the $3+1$d fermions to specific modes, we obtain the massless $2+1$d fermion theory, and the $3+1$d symmetries are reduced to the corresponding $2+1$d symmetries. We can determine whether it is possible to impose a boundary condition preserving the $3+1$d symmetries by using the result of the $2+1$d setup in Sec. 4. The derivation of the inconsistency of the symmetric boundary condition gives a physical interpretation of the Smith homomorphism [16] between the corresponding bordism groups $\Omega_5^{\text{spin}^c}(B\mathbb{Z}_2)$ and $\Omega_4^{\text{pin}^c}$.

## 2 Boundary conditions preserving some non-anomalous symmetries

Let us focus on some spacetime symmetries listed in Tab. 1. We consider $N_f$ free massless Majorana fermions, whose action is given as

$$S = \int d^3x \, \psi_j^T (-i) \gamma^t \gamma^\mu \partial_\mu \psi_j, \tag{1}$$

where $j$ is a flavor index. Here we define the gamma matrices as

$$\gamma^t = i\sigma_2, \qquad \gamma^x = \sigma_1, \qquad \gamma^y = -\sigma_3. \qquad (2)$$

For this choice, the Majorana condition is $\psi = \psi^*$. The theory can have the symmetries and the corresponding anomalies, depending on $N_f$.

By changing the variable as

$$\Psi = \psi_1 + i\psi_2, \qquad (3)$$

we can rewrite the action of two Majorana fermions as the action of one Dirac fermion as

$$\int d^3 x\, \Psi^\dagger (-i)\gamma^t \gamma^\mu \partial_\mu \Psi. \qquad (4)$$

The symmetries act on the Dirac fermion fields $\Psi$ as

$$U(1): \Psi(t,x,y) \to e^{i\alpha}\Psi(t,x,y), \qquad (5)$$
$$R: \quad \Psi(t,x,y) \to \gamma^x \Psi(t,-x,y), \qquad (6)$$
$$\tilde{R}: \quad \Psi(t,x,y) \to i\gamma^x \Psi(t,-x,y), \qquad (7)$$
$$CR: \quad \Psi(t,x,y) \to \gamma^x \Psi^*(t,-x,y). \qquad (8)$$

The symmetries $U(1)$, $\tilde{R}$, and $CR$ are defined only when $N_f$ is even. The reflection $R$ can be defined for any $N_f$, where it acts on the Majorana fermion field $\psi_j$ as

$$\psi(t,x,y) \to \gamma^x \psi(t,-x,y). \qquad (9)$$

The symmetry $\widetilde{CR}$ characterized by $\widetilde{CR}^2 = (-1)^F$ is defined for a pair of Dirac fermions $(\Psi, \widetilde{\Psi})$ as

$$\widetilde{CR}: \quad \Psi(t,x,y) \to \gamma^x \widetilde{\Psi}^*(t,-x,y), \qquad \widetilde{\Psi}(t,x,y) \to -\gamma^x \Psi^*(t,-x,y). \qquad (10)$$

Corresponding to $R, \tilde{R}, CR, \widetilde{CR}$, we can also introduce several types of time-reversal symmetry acting on the fermion field as[1]

$$CT: \Psi(t,x,y) \to \gamma^t \Psi^*(-t,x,y), \qquad (11)$$
$$\widetilde{CT}: \Psi(t,x,y) \to i\gamma^t \Psi^*(-t,x,y), \qquad (12)$$
$$T: \quad \Psi(t,x,y) \to \gamma^t \Psi(-t,x,y), \qquad (13)$$
$$\tilde{T}: \quad \Psi(t,x,y) \to \gamma^t \widetilde{\Psi}(-t,x,y), \qquad \widetilde{\Psi}(t,x,y) \to -\gamma^t \Psi(-t,x,y). \qquad (14)$$

Note that these transformations are antilinear. By performing Wick rotation, these symmetries are analytically continued to $R, \tilde{R}, CR, \widetilde{CR}$ respectively.

The groups in the lower row of Tab. 1 are understood as follows. Each element of the group corresponds to one set of theories with the same anomaly, and the identity element corresponds to non-anomalous theories. The direct product of theories obeys the group multiplication law. For $R$ symmetry, the single massless Majorana fermion theory corresponds to the generator of $\mathbb{Z}_{16}$, and its 16th power, the 16 Majorana fermion theory, has no anomaly. For $U(1) \times R$ symmetry, the single Dirac theory generates the factor $\mathbb{Z}_8$. The other factor $\mathbb{Z}_2$ is generated by a surface theory of a bosonic SPT phase [5]. For $U(1) \rtimes CR$ symmetry, the single Dirac theory generates a factor $\mathbb{Z}_2$. The other factors $\mathbb{Z}_2^2$ correspond to bosonic SPT phases [4]. For

---

[1]The $CT$ transformation can act on a Majorana fermion as $\psi(t,x,y) \to \gamma^t \psi(-t,x,y)$. Due to the antilinearity, there appears the complex conjugate when it acts on a Dirac fermion defined as Eq. (3).

$U(1) \rtimes \widetilde{CR}$ symmetry, the anomaly $\mathbb{Z}_2$ corresponds to a bosonic SPT phase [5]. We will not consider surface theories of bosonic SPT phases in this paper.

Let us check whether we can impose boundary conditions preserving the symmetries. For simplicity, we consider the half of the Minkowski spacetime parametrized by $(t, x, y)$ whose values are taken in $t, x \in \mathbb{R}$ and $y \in \mathbb{R}_{\geq 0}$. The boundary condition should be imposed so that the Hamiltonian $\mathcal{H} := -i\gamma^t \gamma^k \partial_k$ of the one-particle state is Hermitian for the completeness of the eigenfunctions, and thus general spinor fields $\psi, \tilde{\psi}$ should satisfy

$$
\begin{aligned}
0 &= (\tilde{\psi}, \mathcal{H}\psi) - (\mathcal{H}\tilde{\psi}, \psi) \\
&= \int_{-\infty}^{\infty} dx \int_0^{\infty} dy \left[ \tilde{\psi}_j^\dagger (-i)\gamma^t \gamma^k \partial_k \psi_j - ((-i)\gamma^t \gamma^k \partial_k \tilde{\psi}_j)^\dagger \psi_j \right] \\
&= \int_{-\infty}^{\infty} dx \, \tilde{\psi}_j^\dagger (-i)\gamma^t \gamma^y \psi_j \Big|_{y=0}.
\end{aligned}
\tag{15}
$$

A possible linear boundary condition is

$$
\gamma^y \psi_j = M_{jk} \psi_k,
\tag{16}
$$

where $M$ is a real symmetric orthogonal matrix. We see that $R$ is not preserved under this boundary condition, which is consistent with the fact that $R$ has an anomaly. We can impose a boundary condition preserving $U(1)$ using Dirac fields as

$$
\gamma^y \Psi_j = M_{jk} \Psi_k,
\tag{17}
$$

where $M$ is a Hermitian unitary matrix. This is consistent with the fact that $U(1)$ does not have an anomaly. This boundary condition does not preserve $R, \tilde{R}$, and preserves $CR$ only if $M$ is antisymmetric. Since the dimension of an antisymmetric unitary matrix has to be even, $CR$ can be preserved only if the number of the Dirac fermions, $N_f/2$, is even. This is consistent with the fact that $U(1) \rtimes CR$ is non-anomalous only in that case. Instead, the boundary condition

$$
\gamma^y \Psi_j = M_{jk} \Psi_k^*,
\tag{18}
$$

where $M$ is a symmetric unitary matrix,[2] preserves $\tilde{R}$, not $U(1)$ and $R$, and preserves $CR$ only if $M = -M^*$. This is consistent with the fact that $\tilde{R}$ and $CR$ do not have anomalies.[3] At least for the simple boundary conditions (17) and (18), the symmetries $R$ and $U(1) \times R$ are not preserved, which is consistent with the fact that they have anomalies. For $U(1) \rtimes \widetilde{CR}$, there are no anomalies for free fermion fields and we can impose the boundary condition preserving it as

$$
\gamma^y \Psi_j = M_{jk} \Psi_k, \qquad \gamma^y \tilde{\Psi}_j = -M_{jk}^* \tilde{\Psi}_k,
\tag{19}
$$

where $M$ is a Hermitian unitary matrix.

It is known that when there are 16 Majorana fermions (8 Dirac fermions), the reflection symmetry $R$ does not have an 't Hooft anomaly [8–10]. Therefore it is expected that we can

---

[2]Here we define the inner product $(\bullet, \bullet)$ in Eq. (15) using the Majorana fermion fields via the relation (3), i.e., $(\tilde{\Psi}, \Psi) = \int d^3 x (\tilde{\psi}_1^\dagger \psi_1 + \tilde{\psi}_2^\dagger \psi_2)$, which is different from the inner product for the Dirac fields $(\tilde{\Psi}, \Psi) = \int d^3 x \, \tilde{\Psi}^\dagger \Psi$. Naturally, with the boundary condition (17), the Hamiltonian is Hermitian independent of the definition of the inner product. However, with the boundary condition (18), we cannot use the inner product for the Dirac fields, because $\Psi$ satisfying (18) does not span a linear space since the condition (18) is not invariant under the scalar multiplication $\Psi \to i\Psi$.

[3]To understand $CR$ does not have an anomaly, let us decompose a Dirac fermion into two Majorana fermions as Eq. (3). The action of $CR$ is rewritten as $\psi_1(t, x, y) \to \gamma^x \psi_1(t, -x, y)$, $\psi_2(t, x, y) \to -\gamma^x \psi_2(t, -x, y)$, which can be understood as a reflection acting on $\psi_1$ and $\psi_2$ with opposite signs. In this case, the reflection anomalies coming from $\psi_1$ and $\psi_2$ are cancelled, and there are no anomalies in total.

impose a boundary condition preserving $R$ when there are 16 Majorana fermions. However, the simple boundary conditions (17) and (18) do not preserve $R$ irrespective of the number of flavors. In Sec. 4, we show that a non-trivial non-linear boundary condition for 16 Majorana fermions preserves $R$.

## 3 Inconsistency of an $R$ symmetric boundary condition

We impose the following $R$ symmetric boundary condition, which varies at the origin:

$$\gamma_y \psi = -\psi, \quad \text{at} \ y = 0, \ x > 0, \qquad \gamma_y \psi = \psi, \quad \text{at} \ y = 0, \ x < 0. \tag{20}$$

However, this boundary condition should be problematic because $R$ has an anomaly. We will see that an incoming fermion in a specific mode "disappears" at the boundary under this boundary condition. To obtain a corresponding outgoing fermion mode, we need to introduce an additional fermion and impose a boundary condition at the origin.

We can decompose $\psi$ as[4]

$$\psi = \sum_{n=0}^{\infty} \left( \frac{1}{\sqrt{r}} f_n(t,r) \cos(n\theta) \begin{pmatrix} \cos\frac{\theta}{2} \\ \sin\frac{\theta}{2} \end{pmatrix} + \frac{1}{\sqrt{r}} g_n(t,r) \sin(n\theta) \begin{pmatrix} -\sin\frac{\theta}{2} \\ \cos\frac{\theta}{2} \end{pmatrix} \right), \tag{21}$$

where we introduce $r, \theta$ as

$$x = r\cos\theta, \qquad y = r\sin\theta. \tag{22}$$

We can confirm the functions make a complete set as follows. First we decompose with respect to the spinor degrees of freedom as

$$\psi = F(t,r,\theta) \begin{pmatrix} \cos\frac{\theta}{2} \\ \sin\frac{\theta}{2} \end{pmatrix} + G(t,r,\theta) \begin{pmatrix} -\sin\frac{\theta}{2} \\ \cos\frac{\theta}{2} \end{pmatrix}. \tag{23}$$

Due to the boundary condition (20), $G(t,r,\theta)$ has to satisfy $G(t,r,0) = G(t,r,\pi) = 0$, while $F(t,r,\theta)$ is an arbitrary function. Any function $f(\theta)$ of $\theta \in [0,\pi]$ can be decomposed with respect to $\cos(n\theta)$ for $n = 0, 1, \ldots$, because it can be extended to a $2\pi$-periodic even function by defining $f(-\theta) = f(\theta)$ for $\theta \in [0,\pi]$. On the other hand, any function $g(\theta)$ of $\theta \in [0,\pi]$ that is zero at $\theta = 0, \pi$ can be decomposed using $\sin(n\theta)$ because it can be extended to a $2\pi$-periodic odd function by defining $g(-\theta) = -g(\theta)$ for $\theta \in [0,\pi]$. Thus, we obtain the decomposition (21). The Dirac equation $\gamma^\mu \partial_\mu \psi = 0$ implies

$$(\partial_t - \partial_r) f_n - \frac{n}{r} g_n = 0, \qquad (\partial_t + \partial_r) g_n + \frac{n}{r} f_n = 0. \tag{24}$$

A solution with fixed energy $E$ is

$$f_n(t,r;E) = e^{iE(t-r)} r^n M(n+1, 2n+1, 2iEr),$$
$$g_n(t,r;E) = -e^{iE(t+r)} r^n M(n+1, 2n+1, -2iEr), \tag{25}$$

where $M(a,b,x)$ is Kummer's function. To confirm this, we have used the properties of Kummer's function

$$z \frac{dM(a,b,z)}{dz} = (b-a)M(a-1,b,z) + (a-b+z)M(a,b,z),$$
$$M(a,b,z) = e^z M(b-a,b,-z). \tag{26}$$

---

[4]If there is no boundary, we cannot use this decomposition because each component is not $2\pi$ periodic with respect to $\theta$.

For $\psi$ to be real, the coefficient of the positive frequency mode must be the same as that of the negative frequency mode because $M^*(n+1, 2n+1, 2iEr) = M(n+1, 2n+1, -2iEr)$. Asymptotically, the solution (25) behaves as

$$f_n(t, r; E) \sim e^{iE(t+r)}, \qquad g_n(t, r; E) \sim e^{iE(t-r)}. \tag{27}$$

This means that $f_n(t, r; E)$ can be regarded as an incoming wave into the origin, while $g_n(t, r; E)$ can be regarded as the corresponding outgoing wave.

There is a problem with this solution of the Dirac equation. For each mode with $n > 0$, there is an outgoing wave $g_n$ corresponding to the incoming wave $f_n$. However, for $n = 0$, we do not have the outgoing wave because the term containing $g_0$ in Eq. (21) vanishes. This means that an incoming fermion in the $n = 0$ mode has to disappear at the boundary, leading to the violation of the energy conservation.[5] Thus, the reflection symmetric boundary condition (20) is not justified, which is consistent with the fact that the reflection has an anomaly.

To avoid this problem, one can introduce an additional fermion $\tilde{\psi}$ that satisfies the boundary condition

$$\gamma_y \tilde{\psi} = \tilde{\psi}, \quad \text{for } x > 0, \qquad \gamma_y \tilde{\psi} = -\tilde{\psi}, \quad \text{for } x < 0, \tag{28}$$

where the sign of the right-hand side is opposite to the boundary condition (20) for $\psi$. With this boundary condition, there is no incoming wave for $n = 0$ rather than an outgoing wave. We decompose $\tilde{\psi}$ as

$$\tilde{\psi} = \sum_{n=0}^{\infty} \left( \frac{1}{\sqrt{r}} \tilde{f}_n(t, r) \sin(n\theta) \begin{pmatrix} \cos\frac{\theta}{2} \\ \sin\frac{\theta}{2} \end{pmatrix} + \frac{1}{\sqrt{r}} \tilde{g}_n(t, r) \cos(n\theta) \begin{pmatrix} -\sin\frac{\theta}{2} \\ \cos\frac{\theta}{2} \end{pmatrix} \right). \tag{29}$$

Note that the $\sin(n\theta)$ and $\cos(n\theta)$ appear differently as before to satisfy the boundary condition (28). We obtain the same equation as $f_n$ and $g_n$. There is only outgoing mode for $n = 0$ because the $\tilde{f}_0$ term in Eq. (29) vanishes. We can solve the puzzle by letting $\tilde{g}_0$ be the outgoing state corresponding to $f_0$. To achieve this, we impose the boundary condition for the $n = 0$ mode as

$$f_0(t, 0) = \pm \tilde{g}_0(t, 0). \tag{30}$$

This condition can be rewritten as

$$\gamma^t \psi(r = 0) = \mp \tilde{\psi}(r = 0), \tag{31}$$

since only the $n = 0$ modes have nonzero values at the origin. This boundary condition breaks the reflection symmetry (6). Instead, we have a kind of time-reversal symmetry

$$\psi(t) \to \gamma^t \tilde{\psi}(-t), \qquad \tilde{\psi}(t) \to -\gamma^t \psi(-t), \tag{32}$$

which is antilinear. We see that this symmetry is $\widetilde{CT}$ in Eq. (12) by identifying

$$\Psi = \psi + i\tilde{\psi}. \tag{33}$$

Since $\widetilde{CT}$ corresponding to $\tilde{R}$ does not have an anomaly, it is consistent.

# 4 An $R$ symmetric boundary condition for $N_f = 16$

We can impose a boundary condition preserving $R$ when the number of Majorana fermions is a multiple of 16 as follows. Let $\psi_j$ denote half of the Majorana fermions, and $\tilde{\psi}_j$ denote the

---

[5]In other words, the unitarity of the S-matrix is violated. When the initial state is the s-wave, there is no corresponding final state.

**3d Two Majorana** **n=0 mode** **2d Single Majorana**
**Reflection anomaly** ⟹ **Chiral fermion parity anomaly**
$2 \in \mathbb{Z}_{16}$ $1 \in \mathbb{Z}_8$

**Eight copies**

**3d 16 Majorana** ⟹ **2d Eight Majorana**
$0 \in \mathbb{Z}_{16}$ $0 \in \mathbb{Z}_8$

Figure 1: The correspondence between the 3d Majorana fermion theory and 2d Majorana fermion theory. By restricting the fermions to the $n = 0$ modes, the 3d two Majorana fermions $\psi$ and $\tilde{\psi}$ reduce to the 2d single Majorana fermion ($\psi^{2d}, \tilde{\psi}^{2d}$), where $\psi^{2d}$ ($\tilde{\psi}^{2d}$) is the 2d left-moving (right-moving) Weyl fermions. The 3d reflection symmetry reduces the 2d chiral fermion parity. The 3d two Majorana fermion theory has the anomaly corresponding to $2 \in \mathbb{Z}_{16}$. On the other hand, the 2d single Majorana fermion theory has the anomaly corresponding to $1 \in \mathbb{Z}_8$. The eight copies of each theories do not have the anomaly.

rest. We impose the boundary conditions (20) and (28) for $\psi_j$ and $\tilde{\psi}_j$ respectively. Since there is no problem for the modes other than $n = 0$, we can restrict our consideration to the $n = 0$ modes. By substituting the $n = 0$ modes back into the action, we obtain the two-dimensional theory as

$$S_{2d} = \int dt \int_0^{\infty} dr \left[ (\psi_j^{2d} i(\partial_t - \partial_r)\psi_j^{2d} + \tilde{\psi}_j^{2d} i(\partial_t + \partial_r)\tilde{\psi}_j^{2d} \right], \tag{34}$$

where $\psi_j^{2d} = f_{0,j}$ and $\tilde{\psi}_j^{2d} = \tilde{g}_{0,j}$. Here $\psi_j^{2d}$ can be regarded as a left-moving Majorana-Weyl fermion and $\tilde{\psi}_j^{2d}$ as a right-moving one. The reflection $R$, Eq. (6), reduces to the transformation $(\psi_j^{2d}, \tilde{\psi}_j^{2d}) \to (\psi_j^{2d}, -\tilde{\psi}_j^{2d})$, which is the chiral fermion parity. Since the boundary condition at $r = 0$ is only relevant to the $n = 0$ mode, we can describe the condition in the $1 + 1d$ massless fermion theory, where we can use the BCFT language. The symmetry $R$ is preserved if we impose the boundary condition preserving the chiral fermion parity for the 2d fermions $(\psi_j^{2d}, \tilde{\psi}_j^{2d})$.

It is known that when the number of the 2d Majorana fermions is a multiple of 8, the theory does not have the anomaly of the chiral fermion parity. The chiral fermion parity is the internal $\mathbb{Z}_2$ symmetry, whose anomaly is classified by $\mathbb{Z}_8 \times \mathbb{Z}$ because the $\mathbb{Z}_2$ symmetric 3d SPT phase is classified by it [7]. The generating theory of the subgroup $\mathbb{Z}_8$ of the anomaly is the single Majorana fermion theory, and thus the direct product of the eight copies of the theory, the 8 Majorana fermion theory, does not have the anomaly. This is consistent with the fact that the 3d 16 Majorana fermion theory does not have the anomaly of the reflection. See Fig. 1. Consistently with the fact that the theory does not have the anomaly of the chiral fermion parity when the number of the Majorana fermions is a multiple of eight, it is also known that we can impose the boundary condition preserving the chiral fermion parity in such cases [12, 15]. Thus, we can finally conclude that when the number of the Majorana fermion is a multiple of 16, the reflection $R$ is preserved when we impose the boundary condition (20) for half of the fermions and (28) for the others at $y = 0$, while also imposing the boundary condition preserving the chiral fermion parity for the $n = 0$ modes as 2d Majorana fermions at the origin.

When $N_f = 16$, i.e., the number of the 2d Majorana fermions is 8, the 2d boundary condition preserving the chiral fermion parity is the Maldacena-Ludwig boundary condition. The boundary condition is non-linear and defined as the boundary state using open-closed duality. Due to this non-linearity, the outgoing wave corresponding to a usual incoming fermion is an exotic state, which is written as a kink soliton in the 2d bosonized theory [18].

# 5 Relation to the fermion-monopole scattering in 3+1d

As seen before, it is impossible to impose a boundary condition preserving the reflection symmetry for a single Majorana fermion due to the absence of a corresponding outgoing wave for the incoming wave in the $n = 0$ mode. The same issue arises in the context of the scattering between a monopole and a charged fermion in $3 + 1$ dimensions. Before and after the scattering with the monopole, the s-wave component of the fermion has to flip its helicity [19–21], which means that when there is only one Weyl fermion, there is no outgoing wave. This situation can be interpreted as a consequence of the mixed gauge-gravitational anomaly.[6] In this section, we establish a direct connection between the absence of an outgoing s-wave in the monopole scattering in $3 + 1$ dimensions and the impossibility of imposing a boundary condition preserving the reflection symmetry in $2 + 1$ dimensions.

Let us consider the single free left-handed Weyl fermion theory with the unit positive charge in the monopole background,

$$S = \int d^4x \, \chi_L^\dagger i\bar{\sigma}^\mu(\partial_\mu - iA_\mu)\chi_L, \qquad A_\mu = \frac{1}{2}(1 - \cos\theta)\partial_\mu\varphi. \tag{35}$$

In the single Weyl fermion theory, the $U(1)$ symmetry has an 't Hooft anomaly, which means that the theory becomes inconsistent when we introduce a background $U(1)$ gauge field. Therefore, the theory (35) should have an inconsistency. In this case, there is no corresponding outgoing wave for the incoming s-wave, the state with zero angular momentum. In the monopole background, the angular momentum is given as

$$\vec{J} = -i\vec{x} \times (\vec{\nabla} - i\vec{A}) + \frac{1}{2}\vec{\sigma} - \frac{1}{2}\frac{\vec{x}}{r}. \tag{36}$$

Let us decompose the fermion field with respect to the eigenfunctions of $J^2$ and $J_3$. There are two eigenfunctions belonging to the eigenvalues $j(j + 1)$ and $m$ of $J^2$ and $J_3$ as

$$\chi_j^m(\theta, \varphi) = Y_j^m(\theta, \varphi)\begin{pmatrix} \cos\frac{\theta}{2} \\ e^{i\varphi}\sin\frac{\theta}{2} \end{pmatrix}, \qquad \eta_j^m(\theta, \varphi) = \left[\left(\partial_\theta + i\frac{1}{\sin\theta}\partial_\varphi\right)Y_j^m(\theta, \varphi)\right]\begin{pmatrix} -\sin\frac{\theta}{2} \\ e^{i\varphi}\cos\frac{\theta}{2} \end{pmatrix}. \tag{37}$$

When we decompose $\chi_L$ as

$$\chi_L = \sum_{j=0}^\infty \sum_{m=-j}^j \left(\frac{1}{r}f_j^m \chi_j^m + \frac{1}{r}g_j^m \eta_j^m\right), \tag{38}$$

the Dirac equation implies the equations,

$$(\partial_t - \partial_r)f_j^m(t, r) + \frac{j(j+1)}{r}g_j^m(t, r) = 0, \qquad (\partial_t + \partial_r)g_j^m(t, r) - \frac{1}{r}f_j^m(t, r) = 0, \quad \text{for } j > 0,$$
$$(\partial_t - \partial_r)f_0^0(t, r) = 0. \tag{39}$$

Since $\eta_0^0 = 0$, the term containing $g_0^0$ does not appear in the decomposition, and thus there are no outgoing waves for the $j = 0$ mode.

The explicit relation to the $2 + 1$ dimensional system is obtained by restricting the fermion field to the $m = 0$ component. The $m = 0$ component can be expressed as

$$\chi_L = \frac{1}{\sqrt{\rho}}\begin{pmatrix} \Psi_1(t, \rho, z) \\ e^{i\varphi}\Psi_2(t, \rho, z) \end{pmatrix}, \tag{40}$$

---

[6]The correspondence between the mixed gauge-gravitational anomaly and the absence of an outgoing wave can be seen from the fact that they disappear only when the sum of the $U(1)$ charges of the Weyl fermions is zero. See Appendix A.

where the factor $1/\sqrt{\rho}$ is introduced to ensure that the mass dimension of $\Psi_j$ matches that of a $2+1$d fermion. When we restrict the fermion field to this component, the action reduces to

$$\int d^4x \chi_L^\dagger i\bar{\sigma}^\mu(\partial_\mu - iA_\mu)\chi_L = \int dt \int_0^\infty d\rho \int_{-\infty}^\infty dz \left( -\bar{\Psi}\gamma_{3d}^\mu \partial_\mu \Psi - \left(\frac{1}{2\rho} - \frac{1}{\rho}A_\varphi\right)\bar{\Psi}\Psi \right), \quad (41)$$

where we define $\Psi := (\Psi_1, \Psi_2)^T$, $\gamma_{3d}^0 := i\sigma_2$, $\gamma_{3d}^z := \sigma_1$, $\gamma_{3d}^\rho := -\sigma_3$, $\bar{\Psi} := i\Psi^\dagger\gamma^0$. This theory is the 3d fermion coupled with the scalar field

$$\Phi = \frac{1}{2\rho} - \frac{1}{\rho}A_\varphi = \frac{z}{2\rho\sqrt{\rho^2 + z^2}}. \quad (42)$$

Because this scalar field is odd under the reflection $\Phi(\rho, -z) = -\Phi(\rho, z)$, the system has the symmetry under the reflection $\Psi(t, \rho, z) \to \gamma_{3d}^z \Psi(t, \rho, -z)$. Due to the singularity of $\Phi$ at $\rho = 0$, the equation of motion implies the behavior of the fermion fields as

$$\lim_{\rho\to 0} \frac{1}{\sqrt{\rho}}\Psi_2 = 0 \text{ for } z > 0, \qquad \lim_{\rho\to 0} \frac{1}{\sqrt{\rho}}\Psi_1 = 0 \text{ for } z < 0. \quad (43)$$

These behaviors correspond to the reflection symmetric boundary condition (20) by identifying $(x, y)$ as $(z, \rho)$. In $2+1$ dimensions, we can interpret the absence of an outgoing wave in the $j = 0$ mode as a result of this boundary condition preserving the reflection symmetry.

A right-handed Weyl fermion $\chi_R$ with the unit positive charge plays a role of $\tilde{\psi}$ in the previous $2+1$d setup, i.e., it has only the outgoing wave in the $j = 0$ mode. We can decompose $\chi_R$ as

$$\chi_R = \sum_{j=0}^\infty \sum_{m=-j}^j \left(\frac{1}{r}\tilde{g}_j^m \chi_j^m + \frac{1}{r}\tilde{f}_j^m \eta_j^m\right), \quad (44)$$

which is mostly the same as Eq. (38), but the Dirac equation for the right-handed fermion implies $\tilde{g}_j^m$ corresponds to the outgoing wave, and $\tilde{f}_j^m$ corresponds to the incoming one. Thus, the incoming wave $\tilde{f}_0^0$ in the $j = 0$ mode is absent. The $m = 0$ mode is related to the $2+1$d fermion $\tilde{\Psi} = (\tilde{\Psi}_1, \tilde{\Psi}_2)^T$ as

$$\chi_R = \frac{1}{\sqrt{\rho}}\begin{pmatrix} \tilde{\Psi}_2 \\ -e^{i\varphi}\tilde{\Psi}_1 \end{pmatrix}. \quad (45)$$

The reduced $2+1$d theory has the opposite sign of the term containing the scalar field $\Phi$, which implies the behavior near $\rho = 0$ corresponding to the boundary condition (28).

The reflection symmetry of the reduced $2+1$ dimensional fermion $\Psi$ can be extended to $3+1$ dimensions as

$$\chi_L(t, x, y, z) \to e^{i\varphi}\sigma_x \chi_L(t, x, -y, -z), \qquad \chi_R(t, x, y, z) \to -e^{i\varphi}\sigma_x \chi_R(t, x, -y, -z). \quad (46)$$

We can confirm that this reduces to the reflection symmetry by substituting Eqs. (40) and (45). By a rotation and a gauge transformation, this reduces to

$$\chi_L \to \chi_L, \qquad \chi_R \to -\chi_R, \quad (47)$$

which is a chiral rotation. The $U(1)$ background gauge field breaks the $\mathbb{Z}_2$ symmetry (47) due to the chiral anomaly, corresponding to the boundary condition (31) breaking the reflection symmetry in the previous $2+1$d setup.

The four-flavor Dirac fermion theory has exotic outgoing waves that cannot be expressed as Fock states created by a single fermion field when we impose an $SU(4)$ symmetric boundary

**4d Single Dirac** **m=0 mode** **3d Two Dirac** **j=0 mode** **2d Single Dirac**

$U(1) \times \mathbb{Z}_2$ **anomaly** $\Longrightarrow$ $U(1) \times R$ **anomaly** $\Longrightarrow$ $U(1) \times \mathbb{Z}_2$ **anomaly**

$1 \in \mathbb{Z}_8$ $\qquad\qquad$ $2 \in \mathbb{Z}_8$ $\qquad\qquad$ $1 \in \mathbb{Z}_4$

**Four copies**

**4d Four Dirac** $\Longrightarrow$ **3d Eight Dirac** $\Longrightarrow$ **2d Four Dirac**

$4 \in \mathbb{Z}_8$ $\qquad\qquad$ $0 \in \mathbb{Z}_8$ $\qquad\qquad$ $0 \in \mathbb{Z}_4$

Figure 2: The correspondence of the 4d, 3d, and 2d Dirac theories. The 4d theory is considered in the monopole background. By restricting the field to the $m = 0$ modes, whose third component $J_3$ of the angular momentum is zero, the 4d single Dirac fermion $(\chi_L, \chi_R)^T$ reduces to the 3d two Dirac fermions $\Psi$ and $\tilde{\Psi}$ (coupled with the scalar field $\Phi$ odd under the reflection), which obeys the reflection symmetric boundary condition. By further restricting the field to the $j = 0$ mode, the zero angular momentum component, they reduce to the 2d single Dirac fermion. The 4d $U(1) \times \mathbb{Z}_2$ symmetry reduces to the 3d $U(1) \times R$, and it further reduces to the 2d $U(1) \times \mathbb{Z}_2$. The 4d theory in the monopole background cannot detect the anomaly of the 4d four Dirac fermion theory.

condition at the core of the monopole [18, 22–29]. The boundary condition is realized as the Maldacena-Ludwig boundary condition in the two-dimensional theory obtained by restricting the fermion field to the $j = 0$ mode. In this case, the $3 + 1$d chiral fermion parity (47) is preserved, because the Maldacena-Ludwig boundary condition preserves the corresponding 2d chiral fermion parity. This setup corresponds to the $N_f = 16$ Majorana fermion theory in the previous $2 + 1$d setup, where the reflection symmetry can be preserved consistently.

We found that the $3+1$d chiral fermion parity $\mathbb{Z}_2$ given by Eq. (47) is reduced to the reflection symmetry in $2 + 1$ dimensions, and is further reduced to the 2d chiral fermion parity in the monopole background. The monopole background can detect the 4d anomaly of $U(1) \times \mathbb{Z}_2$ that cannot be detected by the chiral anomaly. For two Dirac fermions, $\mathbb{Z}_2$ is not broken by the chiral anomaly, but the scattering with the monopole breaks $\mathbb{Z}_2$, which corresponds to the $R$ symmetry breaking due to the boundary condition in $2 + 1$d four Dirac fermion theory. However, the monopole background cannot detect the anomaly of four Dirac fermions. It is known that the anomaly of the $U(1) \times \mathbb{Z}_2$ is classified by[7] $\mathbb{Z}_8 \times \mathbb{Z}_2$, and the single Dirac fermion theory corresponds to $1 \in \mathbb{Z}_8$, which means that the theory is non-anomalous only if the number of the Dirac fermions is a multiple of eight. See Fig. 2. For the consistency check that the anomaly detected by the $j = 0$ mode is a part of the $3 + 1$d anomaly $\mathbb{Z}_8$ of the $U(1) \times \mathbb{Z}_2$, see Appendix B. We can detect the anomaly $\mathbb{Z}_8$ of $U(1) \times \mathbb{Z}_2$ completely by introducing a boundary condition changing at a line in three spatial dimensions as we will see in the next section.

# 6 A 4d anomaly detected by the 3d reflection anomaly

The classifications of the anomalies of $U(1) \times \mathbb{Z}_2$ in $3 + 1$ dimensions and $U(1) \times R$ in $2 + 1$ dimensions are the same. This coincidence can be explained by the Smith homomorphism, which gives a one-to-one correspondence between corresponding bordism groups [16]. In this section, we see an explicit relation between these $3 + 1$d and $2 + 1$d anomalies using a boundary condition changing at a line similarly to the previous $2 + 1$d setup, where we relate the $2 + 1$d anomaly and the $1 + 1$d anomaly. Unlike the case of the monopole background, this

---

[7]The classification of the anomaly of $U(1) \times \mathbb{Z}_2$ in four dimensions corresponds to the bordism group $\Omega_5^{\text{spin}^c}(B\mathbb{Z}_2) = \mathbb{Z}_8 \times \mathbb{Z}_2$ [30].

method can detect the full 3 + 1d anomaly.

The anomaly of $U(1) \times \mathbb{Z}_2$ can be derived from the anomaly of the $\mathbb{Z}_4$ symmetry whose generator $X$ satisfies $X^2 = (-1)^F$. This anomaly of $\mathbb{Z}_4$ is classified by $\mathbb{Z}_{16}$ [11, 31, 32]. In the following, we initially derive an inconsistency in a $\mathbb{Z}_4$ symmetric boundary condition, and subsequently derive an inconsistency in a $U(1) \times \mathbb{Z}_2$ symmetric boundary condition. Let us consider the single free Weyl fermion theory

$$S = \int d^4x \, \chi_L^\dagger i \bar{\sigma}^\mu \partial_\mu \chi_L \,. \tag{48}$$

We can regard this theory as the single Majorana fermion theory by identifying the Majorana fermion field as $\psi_{4d} = (\chi_L, i\sigma^2 \chi_L^*)^T$. We consider this theory in the region where $x > 0$ and attempt to impose a boundary condition preserving the $\mathbb{Z}_4$ symmetry generated by

$$\psi_{4d} \to -i\gamma^5 \psi_{4d} \quad \Longleftrightarrow \quad \chi_L \to i\chi_L \,, \tag{49}$$

whose square is equal to $(-1)^F$. The simple linear boundary condition $\psi_{4d} = i\gamma_x \psi_{4d}|_{x=0}$ violates this symmetry. Instead, We consider the boundary condition

$$
\begin{aligned}
\psi_{4d} &= -\gamma^5 \gamma^x \psi_{4d} \,, &\text{at } y > 0, \ x = 0 \,, &&\psi_{4d} &= \gamma^5 \gamma^x \psi_{4d} \,, &\text{at } y < 0, \ x = 0 \,, \\
\Longleftrightarrow \quad \chi_L &= -\sigma^3 \chi_L^* \,, &\text{at } y > 0, \ x = 0 \,, &&\chi_L &= \sigma^3 \chi_L^* \,, &\text{at } y < 0, \ x = 0 \,,
\end{aligned} \tag{50}
$$

changing at the $z$ axis. This boundary condition is invariant under the combination of $\mathbb{Z}_4$ and the rotation $(x, y, z) \to (x, -y, -z)$,

$$\psi_{4d}(t, x, y, z) \to -i\gamma^5 \gamma^y \gamma^z \psi_{4d}(t, x, -y, -z) \quad \Longleftrightarrow \quad \chi_L(t, x, y, z) \to \sigma^1 \chi_L(t, x, -y, -z) \,. \tag{51}$$

We will show that this boundary condition presents issues as a result of the pathology of the reflection symmetric boundary condition in a reduced 3d theory.

We decompose the fermion $\chi_L$ as

$$\chi_L = \sum_{n \in \mathbb{Z}} \frac{1}{\sqrt{\rho}} \begin{pmatrix} -i \exp\left(i \frac{2n-1}{2}(\varphi - \frac{\pi}{2})\right) \psi_n^1(t, z, \rho) \\ \exp\left(i \frac{2n+1}{2}(\varphi - \frac{\pi}{2})\right) \psi_n^2(t, z, \rho) \end{pmatrix} \,, \qquad \psi_n^1, \psi_n^2 \in \mathbb{R} \,. \tag{52}$$

It can be shown that any function satisfying the boundary condition (50) can be decomposed in this way as follows. We write $\chi_L = (\chi_L^1, \chi_L^2)^T$. Due to the boundary condition (50), $\chi_L^1 + \chi_L^{1*}$ and $\chi_L^2 - \chi_L^{2*}$ become functions of $\varphi \in [-\pi/2, \pi/2]$ that are zero at $\varphi = \pi/2$. Such a function $f(\varphi)$ can be decomposed by $\sin((2n+1)(\varphi - \pi/2)/2)$ for $n \geq 0$. This is because we can make $f(\varphi)$ be odd under $\varphi \to \pi - \varphi$ and even under $\varphi \to -\pi - \varphi$ by extending the domain to $[-5\pi/2, 3\pi/2]$, which is $4\pi$ periodic function since $f(-5\pi/2) = f(3\pi/2)$. Any $4\pi$ periodic function odd under $\varphi \to \pi - \varphi$ can be decomposed by $\sin(m(\varphi - \pi/2)/2)$, and if $m$ is an odd integer it is even under $\varphi \to -\pi - \varphi$. On the other hand, $\chi_L^1 - \chi_L^{1*}$ and $\chi_L + \chi_L^{2*}$ are functions of $\varphi \in [-\pi/2, \pi/2]$ that are zero at $\varphi = -\pi/2$. In the similar way, we can show that such a function can be decomposed by $\cos((2n+1)(\varphi - \pi/2)/2)$ for $n \geq 0$. Thus, we can write, by using real-valued functions $\chi_n^{\mathrm{Re},1} \chi_n^{\mathrm{Im},1} \chi_n^{\mathrm{Re},2} \chi_n^{\mathrm{Im},2}$ of $t, z, \rho$,

$$
\begin{aligned}
\chi_L^1 + \chi_L^{1*} &= \sum_{n=0}^\infty \sin\left(\frac{2n+1}{2}(\varphi - \pi/2)\right) \frac{\chi_n^{\mathrm{Re},1}}{\sqrt{\rho}} \,, &\qquad \chi_L^1 - \chi_L^{1*} &= \sum_{n=0}^\infty i \cos\left(\frac{2n+1}{2}(\varphi - \pi/2)\right) \frac{\chi_n^{\mathrm{Im},1}}{\sqrt{\rho}} \,, \\
\chi_L^2 + \chi_L^{2*} &= \sum_{n=0}^\infty \cos\left(\frac{2n+1}{2}(\varphi - \pi/2)\right) \frac{\chi_n^{\mathrm{Re},2}}{\sqrt{\rho}} \,, &\qquad \chi_L^2 - \chi_L^{2*} &= \sum_{n=0}^\infty i \sin\left(\frac{2n+1}{2}(\varphi - \pi/2)\right) \frac{\chi_n^{\mathrm{Im},2}}{\sqrt{\rho}} \,.
\end{aligned} \tag{53}
$$

By defining $\psi_n^1, \psi_n^2$ as linear combinations of $\chi_k^{\mathrm{Re},1}$, $\chi_k^{\mathrm{Im},1}$, $\chi_l^{\mathrm{Re},2}$ and $\chi_l^{\mathrm{Im},2}$ properly, we obtain the decomposition (52).

By substituting the decomposition (52) into the action, we obtain

$$S = \sum_{n \in \mathbb{Z}} \int dt dz d\rho \, \psi_n^T (-i) \gamma_{3d}^0 \left( \gamma_{3d}^\mu \partial_\mu + \frac{n}{\rho} \right) \psi_n , \tag{54}$$

where we define $\psi_n := (\psi_n^1, \psi_n^2)$, $\gamma_{3d}^0 := i\sigma_2$, $\gamma_{3d}^z := \sigma_1$, $\gamma_{3d}^\rho := -\sigma_3$. The transformation (51) acts on $\psi_n$ as

$$\psi_n(t,z,\rho) \to (-1)^n \gamma^z \psi_{-n}(t,-z,\rho) . \tag{55}$$

Due to the presence of the ($\rho$-dependent) mass terms $n\bar{\psi}_n \psi_n / \rho$ in the action for the $n \neq 0$ modes, they do not contribute to the anomaly.[8] Only the $n = 0$ mode contributes to the anomaly, and therefore we can restrict the action into $n = 0$ mode,

$$S_0 = \int dt dz d\rho \, \psi^T (-i) \gamma_{3d}^0 \gamma_{3d}^\mu \partial_\mu \psi , \tag{56}$$

where we define $\psi := \psi_0$. This theory is nothing but the single-Majorana fermion theory, and the transformation (51) reduces to $R$. The $R$ symmetric condition (20) causes the problem as shown in Sec. 3, which can be regarded as a problem of the original symmetry (51) in four dimensions. We can impose the $R$ symmetric condition (20) only if the number of the Majorana fermion is a multiple of 16 as shown in Sec. 4, which means that the anomaly $\mathbb{Z}_{16}$ of the $\mathbb{Z}_4$ symmetry is fully detected by the reduced 3d theory.

An inconsistency of a $U(1) \times \mathbb{Z}_2$ symmetric boundary condition is derived as follows. Let us introduce another left-handed Weyl fermion $\tilde{\chi}_L$, and assign the $U(1)$ charges $+1$ and $-1$ to $\chi_L$ and $\tilde{\chi}_L$ respectively in order to avoid the perturbative anomalies. The fermions $\chi_L$ and $\tilde{\chi}_L$ are taken even and odd under $\mathbb{Z}_2$ respectively. The generator (49) of $\mathbb{Z}_4$, which acts on $\chi_L$ and $\tilde{\chi}_L$ in the same way, can be regarded as the element of $U(1) \times \mathbb{Z}_2$ that acts on the fermion fields as

$$\chi_L \to e^{i\pi/2} \chi_L , \qquad \tilde{\chi}_L \to -1 \cdot e^{-i\pi/2} \tilde{\chi}_L . \tag{57}$$

Thus, the $U(1) \times \mathbb{Z}_2$ symmetry is anomalous in this theory. The direct product of eight copies of the theory is free from this anomaly because it consists of 16 Weyl fermions, which is consistent with the classification $\mathbb{Z}_8 \times \mathbb{Z}_2$ of the anomaly of $U(1) \times \mathbb{Z}_2$. Also in this case, we can relate the anomaly to a 3d anomaly. In order to preserve $U(1)$ symmetry, it is necessary to adjust the boundary condition (50) as

$$\chi_L = -\sigma^3 \tilde{\chi}_L^* , \quad \text{at } y > 0, \ x = 0 , \qquad \chi_L = \sigma^3 \tilde{\chi}_L^* , \quad \text{at } y < 0, \ x = 0 . \tag{58}$$

This boundary condition is invariant under the combination of $\mathbb{Z}_2$ and the rotation $(x,y,z) \to (x,-y,-z)$,

$$\chi_L(t,x,y,z) \to i\sigma^1 \chi_L(t,x,-y,-z) , \qquad \tilde{\chi}_L(t,x,y,z) \to -i\sigma^1 \tilde{\chi}_L(t,x,-y,-z) . \tag{59}$$

The linear combinations $\lambda_L := (\chi_L + \tilde{\chi}_L)/2$ and $\tilde{\lambda}_L := -i(\chi_L - \tilde{\chi}_L)/2$ of the fermion fields satisfy the boundary condition (50), and thus they can be decomposed in the same way as Eq. (52). Since $\chi_L = \lambda_L + i\tilde{\lambda}_L$, the decomposition of $\chi_L$ is obtained by replacing the real

---

[8]We can explicitly see that there is no anomaly for $n \neq 0$ mode by redefining $\lambda_n := \psi_n + \psi_{-n}$ and $\tilde{\lambda}_n := \psi_n - \psi_{-n}$, where the transformation reduces to $\lambda_n(z) \to (-1)^n \gamma^z \lambda_n(-z)$ and $\tilde{\lambda}_n(z) \to (-1)^{n+1} \gamma^z \tilde{\lambda}_n(-z)$, and thus the anomalies coming from $\lambda_n$ and $\tilde{\lambda}_n$ are cancelled.

fields $\psi_n^1, \psi_n^2$ by complex fields $\Psi_n^1, \Psi_n^2$. The other field $\tilde{\chi}_L$ can also be decomposed using the same complex fields $\Psi_n^1, \Psi_n^2$. Thus, the 3d theory obtained by restricting the fields to the $n = 0$ component is the single Dirac fermion theory. The symmetry $U(1)$ and Eq. (59) are reduced to $U(1) \times R$ in this 3d theory. Thus, the anomaly $\mathbb{Z}_8$ of $U(1) \times \mathbb{Z}_2$ is also detected by the reduced 3d theory.

# 7  Summary and discussion

We investigate the boundary conditions (20) and (28) preserving the reflection symmetry $R$ of free massless Majorana fermions. When there are odd number of Majorana fermions, they are problematic, i.e., the condition (20) (resp. (28)) results in the absence of an outgoing (resp. incoming) wave for the $n = 0$ mode in the decomposition (21) (resp. (29)). This is understood as a consequence of the anomaly of $R$. In the case of an even number of Majorana fermions, the problem can be solved by imposing boundary conditions at the origin, such as Eq. (31), where $\psi$ obeying the condition (20) and $\tilde{\psi}$ obeying the condition (28) are mixed. The boundary condition at the origin can be described using BCFT language in $1 + 1$ dimensions. This is possible because the only relevant mode is the $n = 0$ mode, and the theory reduces $1 + 1$ dimensions when the field is restricted to the mode. We conclude that we can impose an $R$ symmetric boundary condition at the origin only if the number of the Majorana fermions are multiple of 16 using the result of $1 + 1$ dimensional CFTs.

In order to preserve $R$ symmetry, it is necessary to impose a nonlinear boundary condition, where the final state is exotic state that cannot be created by a single fermion field. The corresponding state in $1 + 1$ dimensions can be understood as a fractional kink using the bosonization [18, 22]. However, the comprehension in $2 + 1$ dimensions is currently inadequate, and it remains uncertain whether a particle interpretation is applicable. The same problem was considered in the fermion-monopole scattering in 4d, but there is no consensus on the interpretation of the state [18,22–28]. The 2+1d framework might be more convenient for describing the exotic outgoing state than $3 + 1$d.

In this paper, we do not consider the part of the classification of the anomalies related to bosonic SPT phases. It is interesting to consider a nontrivial cancellation of anomalies using a free fermion theory and a bosonic theory. It is known that the 8 Majorana fermion theory in three dimensions can be a surface theory of a bosonic SPT in four dimensions, and the anomaly can be cancelled by a bosonic theory, a $\mathbb{Z}_2$ gauge theory coupled with the Stiefel-Whitney class in a specific way. A boundary condition preserving $R$ in the combined theory of these two could be more exotic.

# Acknowledgements

**Funding information**    This work is partially supported by the National Science and Technology Council (NSTC) of Taiwan under grant numbers 111-2112-M-002 -017- (JWC), 112-2112-M-002 -027 - (JWC), 112-2112-M-002 -048 -MY3 (CTH), and 111-2811-M-002 -051- (RM). CTH is supported by the Yushan (Young) Scholar Program of the Ministry of Education in Taiwan under grant NTU-111VV016.

# A The scattering of a monopole and fermions with general charges

In this section, we show that, in the monopole background, we can find the corresponding incoming (outgoing) wave to any outgoing (incoming) wave only if the sum of the $U(1)$ charges of the Weyl fermions is zero. This condition coincides with the cancellation condition of the mixed gauge-gravitational anomaly.

In the monopole background, for a fermion with charge $Q > 0$, only incoming waves are present among the lowest partial waves. The lowest partial waves are the eigenfunction of $J^2$ belonging to $j(j + 1)$ for $j = (Q - 1)/2$, and thus there are $Q$ independent modes. On the other hand, for the fermion with charge $Q < 0$, only outgoing waves are present among the lowest partial waves, where the number of the independent modes is $-Q$. Thus, if and only if the sum of the charges $\sum_j Q_j$ is zero, the number of the incoming waves and the outgoing waves are the same, and we can set the boundary condition relating them so that the energy is conserved.

In the following, we show, for the sake of completeness, that the lowest partial waves only have incoming modes for $Q > 0$ and outgoing modes for $Q < 0$. This fact was established by the analysis given in Ref. [21]. We consider the left-handed Weyl fermions $\chi^Q$ in the monopole background with the charges $Q$, whose equation of motion is given as

$$\bar{\sigma}^\mu(\partial_\mu - iQA_\mu)\chi^Q = 0, \qquad A_\mu = \frac{1}{2}(1 - \cos\theta)\partial_\mu\varphi. \tag{A.1}$$

The angular momentum is given as

$$\vec{J}^Q = -i\vec{x} \times (\vec{\nabla} - iQ\vec{A}) + \frac{1}{2}\vec{\sigma} - \frac{Q}{2}\frac{\vec{x}}{r}. \tag{A.2}$$

The ladder operators are given as

$$J_\pm^Q := J_1^Q \pm iJ_2^Q = e^{\pm i\varphi}(\pm\partial_\theta + i\cot\theta\,\partial_\varphi) + \frac{Q}{2}e^{\pm i\varphi}\frac{\cos\theta - 1}{\sin\theta} + \frac{1}{2}(\sigma^1 \pm i\sigma^2). \tag{A.3}$$

Let us consider a fermion with a positive charge $Q > 0$. The solution for a fermion with a negative charge $-Q$ is obtained as $i\sigma^2\chi^*(-t, \vec{x})$ using the solution $\chi(t, \vec{x})$ for the fermion with a positive charge $Q$ since

$$\bar{\sigma}^\mu\left(\partial_\mu - i(-Q)A_\mu\right)(i\sigma^2\chi^*(-t, \vec{x})) = -i\sigma^2\left((\partial_{\tilde{t}} - \vec{\sigma}\cdot(\vec{\nabla} - iQ\vec{A}))\chi(\tilde{t}, \vec{x})\right)^*\big|_{\tilde{t}=-t} = 0. \tag{A.4}$$

The function that is an eigenstate of $(J^Q)^2$ and the lowest eigenfunction of $J_3^Q$ is given as

$$\chi_j^{Q,-j} = \left(\cos\frac{\theta}{2}\right)^{Q-1}\chi_{j-(Q-1)/2}^{-(j-(Q-1)/2)}(\theta, \varphi), \qquad \eta_j^{Q,-j} = \left(\cos\frac{\theta}{2}\right)^{Q-1}\eta_{j-(Q-1)/2}^{-(j-(Q-1)/2)}(\theta, \varphi), \tag{A.5}$$

where $\chi_j^m, \eta_j^m$ are defined in Eq. (37). We can confirm that these functions vanish when $J_-^Q$ acts on them by using

$$J_-^Q\left(\cos\frac{\theta}{2}\right)^{Q-1} = \left(\cos\frac{\theta}{2}\right)^{Q-1}J_-^{Q=1}. \tag{A.6}$$

We can also check that these are eigenfunction of $J_3^Q$ belonging to $-j$ by acting

$$J_3^Q = -i\partial_\varphi - \frac{1}{2}Q + \frac{1}{2}\sigma^3. \tag{A.7}$$

Note that the lowest eigenfunction of $(J^Q)^2$ corresponds to $j = (Q-1)/2$, and for this value $\eta_j^{Q,-j} = 0$. The higher eigenfunctions $\chi_j^{Q,m}$, $\eta_j^{Q,m}$ of $J_3^Q$ are obtained by acting $J_+^Q$ repeatedly. We expand the fermion field $\chi^Q$ with charge $Q$ as

$$\chi^Q = \sum_{j=(Q-1)/2}^{\infty} \sum_{m=-j}^{j} \left( \frac{1}{r} f_j^{Q,m}(t,r) \chi_j^{Q,m}(\theta,\varphi) + \frac{1}{r} g_j^{Q,m}(t,r) \eta_j^{Q,m}(\theta\varphi) \right). \tag{A.8}$$

Because the Dirac operator commutes with $J_+^Q$, the equation for each mode does not depend on $m$, and therefore it is enough to consider the lowest eigenfunctions (A.5). By using the properties

$$(-\partial_\theta + i\cot\theta\,\partial_\varphi) Y_{j-(Q-1)/2}^{-(j-(Q-1)/2)} = 0, \qquad -i\partial_\varphi Y_{j-(Q-1)/2}^{-(j-(Q-1)/2)} = -(j-(Q-1)/2) Y_{j-(Q-1)/2}^{-(j-(Q-1)/2)}, \tag{A.9}$$

we obtain the equations for the modes $j > (Q-1)/2$ as

$$(\partial_t - \partial_r) f_j^{Q,m}(t,r) + \frac{4j(j+1) - (Q-1)(Q+1)}{4r} g_j^{Q,m}(t,r) = 0,$$

$$(\partial_t + \partial_r) g_j^{Q,m}(t,r) - \frac{1}{r} f_j^{Q,m}(t,r) = 0, \tag{A.10}$$

and those for the lowest $j$ modes as

$$(\partial_t - \partial_r) f_{(Q-1)/2}^{Q,m}(t,r) = 0. \tag{A.11}$$

We see that there are only incoming waves in the lowest $j$ modes, and no corresponding outgoing waves. The number of the lowest $j$ mode is $Q$. On the other hand, the solution for the fermion with charge $-Q$ is obtained by acting $\chi(t,\vec{x}) \to i\sigma^2 \chi^*(-t,\vec{x})$ to Eq. (A.8) as

$$\chi^{-Q} = \sum_{j=(Q-1)/2}^{\infty} \sum_{m=-j}^{j} \left( \frac{1}{r} g_j^{-Q,m}(t,r) i\sigma^2 \chi_j^{Q,m*}(\theta,\varphi) + \frac{1}{r} f_j^{-Q,m}(t,r) i\sigma^2 \eta_j^{Q,m*}(\theta\varphi) \right),$$

$$g_j^{-Q,m}(t,r) = f_j^{Q,m}(-t,r), \qquad f_j^{-Q,m}(t,r) = g_j^{Q,m}(-t,r). \tag{A.12}$$

Thus, the lowest $j$ modes $g_{(Q-1)/2}^{-Q,m}$ are the outgoing waves.

## B  A check that the anomaly of the s-waves is a part of the anomaly of U(1)×$\mathbb{Z}_2$

In this section, we give a consistency check that the anomaly detected by the s-waves is part of the 4d anomaly $\mathbb{Z}_8$ of the $U(1) \times \mathbb{Z}_2$ symmetry and not part of the other anomaly. We consider left-handed Weyl fermions $\chi_k$ with the $U(1)$ charge $Q_k$, which transforms under $\mathbb{Z}_2$ as $\chi_k \to (-1)^{n_k} \chi_k$ with $n_k \in \{0,1\}$, and determine how the classification of the anomaly depends on $Q_k$ and $n_k$. Then we determine how the classification of the 2d anomaly of the corresponding s-wave theory depends on $Q_k$ and $n_k$. By comparing them, we will conclude that the 2d anomaly of the s-wave theory is part of the 4d anomaly.

### B.1  The anomaly $\mathbb{Z}_8$ of the $U(1) \times \mathbb{Z}_2$ symmetry in four dimensions

The anomaly $\mathbb{Z}_8$ of the $U(1) \times \mathbb{Z}_2$ symmetry is derived from the anomaly $\mathbb{Z}_{16}$ of the $\mathbb{Z}_4$ symmetry whose generator $X$ satisfies $X^2 = (-1)^F$. This $\mathbb{Z}_{16}$ anomaly is considered, e.g., in Refs. [11,31,

32]. A generating theory of this anomaly is the single left-handed Weyl fermion theory, where the generator of $\mathbb{Z}_4$ acts on the fermion as a multiplication by $i$. The direct product of 16 copies of this theory does not have the anomaly. Because the square of the generator of $\mathbb{Z}_4$ has to be equal to $(-1)^F$, the generator has to act on any fermion as a multiplication by $i$ or $-i$. We define $m_k \in \{-1, 1\}$ for each Weyl fermion $\chi_k$ so that the generator of $\mathbb{Z}_4$ acts as $\chi_k \to i^{m_k} \chi_k$. A pair $\chi_k, \chi_l$ of Weyl fermions with $m_k = 1$ and $m_l = -1$ does not contribute to the anomaly because this $\mathbb{Z}_4$ is subgroup of the non-anomalous $U(1)$, $\chi_k \to e^{i\theta} \chi_k$, $\chi_l \to e^{-i\theta} \chi_l$. Thus, the classification of this anomaly is given as[9] $\sum_k m_k$ mod 16, the reduction modulo 16 of the difference of the number of the Weyl fermions with $m_k = 1$ and those of $m_k = -1$.

Let us relate this anomaly to the anomaly $\mathbb{Z}_8$ of $U(1) \times \mathbb{Z}_2$. Precisely speaking, the anomaly $\mathbb{Z}_8$ is the anomaly of the symmetry $\mathrm{Spin}^c(4) \times \mathbb{Z}_2$, where $\mathrm{Spin}^c(4)$ is defined as

$$\mathrm{Spin}^c(4) = \frac{\mathrm{Spin}(4) \times U(1)}{\mathbb{Z}_2}. \tag{B.1}$$

Here $\mathrm{Spin}(4)$ is (the Euclidean version) of the Lorentz symmetry, and the division by $\mathbb{Z}_2$ means that we identify $(-1)^F \in \mathrm{Spin}(4)$ and $-1 \in U(1)$. In order to make $(-1)^F = -1 \in U(1)$, the charges $Q_k$ have to be odd integers. Additionally, in order to avoid the mixed gravitational-gauge anomaly, the sum of the charges has to be zero, $\sum_k Q_k = 0$. In this theory, the $\mathbb{Z}_4$ symmetry stated above is reduced to $\mathbb{Z}_2$ by multiplying $i \in U(1)$,

$$\chi_k \xrightarrow{\mathbb{Z}_4} i^{m_k} \chi_k \xrightarrow{i \in U(1)} i^{Q_k + m_k} \chi_k = (-1)^{(Q_k + m_k)/2} \chi_k, \tag{B.2}$$

where $(Q_k + m_k)/2$ is an integer because both of $Q_k$ and $m_k$ are odd integers. Thus, the anomaly of $\mathbb{Z}_4$ can be regarded as the anomaly of $U(1) \times \mathbb{Z}_2$ in this theory. However, the classification changes from $\mathbb{Z}_{16}$ to $\mathbb{Z}_8$. Because $\sum_k Q_k = 0$ and every $Q_k$ is odd, the number of the Weyl fermions is an even integer, and therefore $\sum_k m_k$ is an even integer. Thus, the anomaly is classified by $\sum_k m_k / 2$ mod 8. We can determine $m_k$ from the charge $Q_k$ and the $\mathbb{Z}_2$ charge $n_k$ as

$$m_k = \begin{cases} 1, & \text{if } n_k = 0, \, Q_k = 3 \bmod 4, \quad \text{or} \quad n_k = 1, \, Q_k = 1 \bmod 4, \\ -1, & \text{if } n_k = 0, \, Q_k = 1 \bmod 4, \quad \text{or} \quad n_k = 1, \, Q_k = 3 \bmod 4. \end{cases} \tag{B.3}$$

## B.2 The anomaly $\mathbb{Z}_4$ of $U(1) \times \mathbb{Z}_2$ in the 2d s-wave theory

As we show in Appendix A, a 4d Weyl fermion $\chi_k$ with charge $Q_k > 0$ (resp. $Q_k < 0$) reduces to $Q_k$ (resp. $-Q_k$) 2d left-moving (resp. right-moving) fermions in the s-wave theory. The $U(1)$ and $\mathbb{Z}_2$ act on these 2d fermions in the same way as the corresponding 4d Weyl fermions. Let us determine the dependence of the anomaly $\mathbb{Z}_4$ of $U(1) \times \mathbb{Z}_2$ on $Q_k$ and $n_k$ in the 2d s-wave theory.

The 2d anomaly $\mathbb{Z}_4$ of the $U(1) \times \mathbb{Z}_2$ symmetry is derived from the 2d anomaly $\mathbb{Z}_8$ of the $\mathbb{Z}_2$ symmetry. The classification $\mathbb{Z}_8$ of the anomaly of $\mathbb{Z}_2$ is written, e.g., in Ref. [7] as the classification of the 3d fermionic SPT phases with the symmetry $\mathbb{Z}_2$. A generating theory of the anomaly $\mathbb{Z}_8$ of $\mathbb{Z}_2$ is the single Majorana fermion theory, where the left-moving fermion is odd and right-moving fermion is even under $\mathbb{Z}_2$. A pair of Majorana fermions whose $\mathbb{Z}_2$ charge assignments are opposite to each other does not have the anomaly, because this $\mathbb{Z}_2$ action becomes $\psi \to -\psi$ for the Majorana fermion $\psi$ made by the left-moving and right-moving components that are odd under $\mathbb{Z}_2$. Therefore the anomaly is classified by the reduction modulo eight of the difference between the number of left-moving and right-moving real fermions that are odd under $\mathbb{Z}_2$. Similar to the previous 4d case, to introduce the $U(1) \times \mathbb{Z}_2$ symmetry,

---

[9]The classification in Ref. [31] reduces to this by choosing the representative of the charges as $m_k \in \{-1, 1\}$.

all fermions must have odd charges. Consequently the number of the Majorana fermions in the theory has to be even, allowing all Majorana fermions to pair up and form Dirac fermions. Therefore the anomaly $\mathbb{Z}_8$ reduces to $\mathbb{Z}_4$. Thus, we conclude that the anomaly $\mathbb{Z}_4$ of $U(1) \times \mathbb{Z}_2$ is classified by the reduction modulo four of the difference between the number of left-moving and right-moving complex fermions that are odd under $\mathbb{Z}_2$.

In the s-wave theory, the number of the left-moving fermions that are odd under $\mathbb{Z}_2$ is $\sum_{Q_k > 0} n_k Q_k$ and the number of the right-moving fermions that are odd is $-\sum_{Q_k < 0} n_k Q_k$, and thus the anomaly $\mathbb{Z}_4$ of $U(1) \times \mathbb{Z}_2$ is classified by $\sum_k n_k Q_k \bmod 4$. Let $l_k \in \{-1, 1\}$ be $l_k = Q_k \bmod 4$. Using this, the classification is given by $\sum_k n_k l_k \bmod 4$. Since $\sum_k Q_k = 0$ and consequently $\sum_k l_k = 0$, the classification can be expressed as $\sum_k (2n_k - 1)l_k/2 \bmod 4$. As $(2n_k - 1)l_k = m_k$, where $m_k$ is defined in Eq. (B.3), we finally arrive at the classification $\sum_k m_k/2 \bmod 4$.

### B.3  Comparing the 4d anomaly and 2d anomaly of the s-wave theory

The anomaly $\mathbb{Z}_8$ of the 4d theory and the anomaly $\mathbb{Z}_4$ of the 2d s-wave theory are controlled by the same quantity $\sum_k m_k/2$, which means that the anomaly $\mathbb{Z}_4$ is a part of the anomaly $\mathbb{Z}_8$ in the sense that we cannot make a theory without the anomaly $\mathbb{Z}_8$ while maintaining the anomaly $\mathbb{Z}_4$. The 4d theory with $\sum_k m_k = 4 \bmod 8$ corresponds to $4 \in \mathbb{Z}_8$ of the classification of the 4d anomaly, but its 2d s-wave theory corresponds to $0 \in \mathbb{Z}_4$ of the classification of the 2d anomaly, i.e., does not have the anomaly. Thus, we conclude that the s-wave theory only detects the reduction modulo four of the full anomaly $\mathbb{Z}_8$.

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
