# Peer review of "Boundary condition and reflection anomaly in $2+1$ dimensions"

_SciPost Physics, doi:SciPost Phys. 17, 068 (2024)_

## Round 2 · Referee Report · Anonymous (Referee 1) · 2024-5-21

Report

This paper explores the possible connection between two notions. The first notion is the existence of a 't Hooft anomaly for a symmetry group G in a quantum field theory. The second notion, dubbed edge-ability, is the existence of G-preserving boundary conditions. The authors explore this theme using the specific example of Majorana fermions in 2+1 dimensions, where the symmetry G in question is reflection. By considering a boundary condition varying at a point they conclude that indeed the existence of the reflection anomaly leads to an inconsistency in reflection symmetric boundary conditions. They further make connections between this lack of edge-ability and recent progress on the fermion-monopole scattering in 3+1 dimensions.

The paper is well written, the logic is clearly explained and the main results are put front and centre away from any superfluous distractions. The question they pose is novel and unexplored in higher dimensions, and the connection they point out to fermion-monopole scattering may have further applications. I therefore think that it can be published in Scipost physics, provided the following points are addressed.

Requested changes

1) The boundary condition in eq (3.1) implies that \psi=0 at the origin (r=0). This is clearly not the case for the 0-mode of the solution they propose in (3.6) since f_0 contains 0^0=1 and moreover the Kummer's function give 1 at r=0. How is this consistent with the boundary condition?

2) Below equation (5.1) the authors claim that coupling a symmetry with a 't Hooft anomaly to a background leads to an inconsistency. This is not true, the theory is perfectly consistent i.e. there is no tension with regularisation as opposed to dynamical symmetries. I ask the authors to rephrase this paragraph appropriately.

Recommendation

Ask for minor revision

  • validity: good
  • significance: high
  • originality: high
  • clarity: top
  • formatting: excellent
  • grammar: excellent

Author:  Ryutaro Matsudo  on 2024-06-05  [id 4539]

(in reply to Report 1 on 2024-05-21)
Category:
answer to question

Thank you very much for reading our manuscript carefully, and giving positive comments. The following are the answers to your questions:

(1) The equation (3.1) does not specify the value of $\psi$ at $r=0$, and only specifies the values of the upper component of $\psi$ at $x< 0$ and the lower component at $x>0$. Also, any function in the expansion (3.2) satisfies the boundary condition, as explained in the paper. You may worry about the fact that the zero mode diverges at r=0, but it does not cause any problem since the action for the $n=0$ modes is just a 2d fermion theory (on a half line), which is finite and gives a consistent theory (when the numbers of the left and right movers are matched). As we explained in the paper, we need to specify additional boundary conditions at r=0 to fix the behavior of the $n=0$ modes.

(2) The single Weyl fermion theory in the monopole background is inconsistent since the corresponding outgoing wave to the s-wave component does not exist. We refer to a theory that cannot preserve energy as an inconsistent theory throughout this paper.

In general, the theory with an 't Hooft anomaly can be inconsistent when we introduce background gauge fields. For example, in the single Weyl fermion theory, when we introduce a background U(1) gauge field with non-vanishing $\vec E\cdot \vec B$, the U(1) symmetry is broken, i.e., the gauge symmetry is broken. A theory is said to be inconsistent when a gauge symmetry is broken. Note that even when we introduce a background gauge field and not a dynamical gauge field, the corresponding symmetry becomes a gauge symmetry. In other words, we cannot introduce a background gauge field when the symmetry is broken.

In the literature, 't Hooft anomalies refer to impossibility to couple background gauge fields. See, e.g., the first paragraph of [D. Gaiotto et al., "Theta, time reversal and temperature", JHEP 05 (2017) 91].

---

## Round 2 · Referee Report · Anonymous (Referee 2) · 2024-7-14

Strengths

1- Very clearly written

2- Novel answer to an old question, to the best of my knowledge. Very little is known about d>2, so this is definitely a significant step forward

3- Same method could potentially be applied to new problems, whose answer is not known

Weaknesses

1- Unclear if the analysis can be used in other contexts or it only works for the specific example studied in the paper (namely free fermions)

2- No new predictions, so hard to say if the argument works in complete generality, or it works here only because the correct answer was already known a priori

3- Relies on the d=2 result of Maldacena-Ludwig instead of providing an independent argument (this complaint applies to most papers on the subject rather than just this work)

Report

The paper tackles a very important question and the answer is compelling, interesting and well presented. I only have minor comments.

1) eqs 2.3--2.13 only work when Nf is even. It could be useful to write e.g. how R acts on Majorana fermions directly instead of Diracs.

2) is Nf the number of Majoranas or Diracs? on page 3 it seems the former but below 2.16 it seems the latter.

3) In general, a sharp discontinuity in the fields might be ambiguous, scheme-dependent, so it could be better to impose a smooth transition and consider a limit. This might shed some light on the fate of the missing mode n=0. That being said, for free theories the ambiguity is most likely absent so no issues arise in this paper.

4) the language around 3.9 is not ideal. introducing an additional fermion is indeed one way to solve the problem, but certainly not the unique one, so "...it is necessary to introduce" is not correct. It would be better to say something like "To avoid this problem, one can introduce..."

Requested changes

It could be useful to make some minor adjustments addressing the points in the report above, but this is not necessary for acceptance.

Recommendation

Publish (meets expectations and criteria for this Journal)

  • validity: good
  • significance: good
  • originality: high
  • clarity: high
  • formatting: excellent
  • grammar: perfect

Author:  Ryutaro Matsudo  on 2024-07-23  [id 4643]

(in reply to Report 2 on 2024-07-14)
Category:
correction

Thank you for reading the manuscript carefully and providing helpful comments. Below are our responses to your comments and the modifications we plan to make in the revised manuscript. 1) We will add a text below (2.8) to clarify the action of the reflection to a Majorana fermion. 2) Thank you for pointing out the misuse of the symbol $N_f$. We will modify the text below Eq. (2.17) to clarify that the number here refers to the number of Dirac fermions, i.e., $N_f/2$. 3) Thank you for the suggestion. It would be interesting to consider a way to make the boundary condition smooth. 4) We will modify the sentence above Eq. (3.9) as you suggested.

---

## Round 3 · List of Changes

1) We added Eq.~(2.9) and sentences above it to explicitly indicate the action of the reflection on a Majorana fermion field.
2) We fixed the misuse of the symbol $N_f$ below the Eq. (2.17).
3) We corrected the sentence above (3.9).

---

## Editorial Decision

published